# Deleterious Effects of Yoyo Dieting and Resistant Starch on Gastrointestinal Morphology

**DOI:** 10.3390/nu16234216

**Published:** 2024-12-06

**Authors:** Kate Phuong-Nguyen, Malik Mahmood, Leni Rivera

**Affiliations:** 1School of Medicine, Institute for Mental and Physical Health and Clinical Translation, Deakin University, Geelong, VIC 3220, Australia; kate.nguyen@research.deakin.edu.au; 2School of Medicine, Deakin University, Geelong, VIC 3216, Australia; malik.mahmood@deakin.edu.au

**Keywords:** resistant starch, yoyo dieting, weight cycling, gastrointestinal structures, inflammation

## Abstract

Background: Obesity is associated with structural deterioration in the gut. Yoyo dieting, which refers to repeated phases of dieting and non-dieting periods leading to cyclic weight loss and regain, is a common occurrence in individuals with obesity. However, there is limited evidence on how gut structures are affected in yoyo dieting. There is good evidence suggesting that increased intake of resistant starch (RS) may be beneficial in promoting structural improvements in the gut. This investigation aimed to explore the effect of yoyo dieting on gastrointestinal structure and whether RS has beneficial effects in improving obesity-related gastrointestinal damage. Method: In this study, male and female C57BL/6 mice were assigned to six different diets for 20 weeks: (1) control diet, (2) high fat diet (HF), (3) yoyo diet (alternating HF and control diets every 5 weeks), (4) control diet with RS, (5) HF with RS, and (6) yoyo diet with RS. Distal colon was collected for epithelial barrier integrity measurement. The small and large intestines were collected for histological assessment. Results: After 20 weeks, yoyo dieting resulted in increased colonic inflammation and exacerbated mucosal damage in comparison with continuous HF diet feeding. RS supplemented in HF and yoyo diets reduced mucosal damage in comparison to diets without RS. However, RS supplementation in a control diet significantly increased inflammation, crypt length, and goblet cell density. There were no significant differences in epithelial change and epithelial barrier integrity across diet groups. Conclusions: This study suggests that yoyo dieting worsens gut damage, and incorporating high levels of RS may be detrimental in the absence of dietary challenge.

## 1. Introduction

Obesity has been recognised as the second leading cause of disease burden in Australia [1] and alarmingly an escalating global epidemic in the modern world [2]. For decades, obesity has been on the rise, with a global prevalence of 38% in 2020 [2,3] and is predicted to surpass 50% by 2035 [2,4,5]. This represents a significant risk factor for a range of metabolic diseases [6,7,8,9,10], such as type 2 diabetes [11], fatty liver [12,13], and cardiovascular diseases [14,15].

Despite the importance of weight loss in mitigating these risks [16,17], achieving sustainable weight loss remains elusive for many individuals due to rapid weight regain [16,17]. Evidence suggests that 80–95% of individuals with obesity regain most or all their lost weight within 5 years [18,19,20,21,22,23,24,25,26,27,28,29,30,31,32,33,34,35]. This phenomenon is called yoyo dieting, which refers to cyclic weight loss and weight regain [36]. Yoyo dieting can undermine the desired outcome of weight loss [24,37,38,39,40], potentially predisposing individuals to further weight gain and exacerbating obesity-related comorbidities [28,41].

Alterations in the gut structure are increasingly implicated in many disease progression [42]. Of note, chronic gut inflammation is a hallmark of obesity [43,44,45,46,47] and gut structural damage is known as an important contributor to obesity-related gastrointestinal dysfunction [48,49]. While there has been good progress in studying the impact of obesity on metabolic health [50], little is known about how yoyo dieting potentially affects intestinal integrity and overall gut health. Limited evidence suggests that yoyo dieting-induced obesity could be responsible for intestinal barrier defects, such as reduction in mucus production, increased intestinal permeability, and disruption of epithelial tight junctions, as well as elevated local and systemic intestinal inflammation [51,52].

Increasing resistant starch (RS) intake has been shown to improve gut health [53,54,55,56,57,58,59,60] and weight management [61,62,63,64,65,66,67]. RS, a type of dietary fibre, is a small portion of starch that cannot be digested in the small intestine and, hence, is passed to the colon to be fermented by the colonic microbiota [66,68,69]. The fermentation of colonic microbiota leads to the production of short-chain fatty acids (SCFAs). These SCFAs are known to provide an energy supply for the colonic mucosa [70,71,72] and act as signalling molecules regulating metabolic pathways [73,74].

Therefore, in the present work, we used a mouse model of yoyo dieting to examine how weight cycling affects gastrointestinal structure and function. This investigation aimed to explore how yoyo dieting affects gastrointestinal structure and explore whether RS supplementation might be a promising nutritional approach to improve gut health.

## 2. Materials and Methods

### 2.1. Animal and Diets

All animal procedures were performed according to the guidelines of the Deakin University Animal Ethics Committee and the National Health and Medical Research Council.

To study the mechanism of yoyo dieting and the potential effects of resistant starch on gut health, male and female C57BL/6 (*n* = 47 males and *n* = 46 females; 5-week-old) were purchased from the Animal Resource Centre (Western Australia, Australia). Following arrival, mice were acclimatised for one week and fed a normal chow diet. At 6 weeks of age, mice were exposed to six diet treatments (*n* = 7–8/diet group, 4 per cage, 2 cages per group) for 20 weeks (Figure 1). Animals were given free access to irradiated food and autoclaved water and were maintained on a strict 12 h on/off light/dark cycle, controlled temperature at 21 °C, and humidity between 40 and 70%. Mice were randomly assigned to 6 dietary groups, including a control diet (Control; SF17-091; 13% kcal from fat), high fat diet (HF; SF16-048; 60% kcal from fat), yoyo diet (Yoyo) in which mice were fed a HF diet interspaced with a control diet every 5 weeks, HF diet supplemented with resistant starch (HF RS; SF21-018; 60% kcal from fat), control diet supplemented with RS (Control RS; SF21-019; 13% kcal from fat), and yoyo diet with RS supplemented in the 2nd weight gain cycle, in which mice were fed a HF diet for 5 weeks, control diet for 5 weeks, HF RS diet for 5 weeks, and control diet for 5 weeks (Yoyo RS) (Figure 1). All diets were customised by Specialty Feeds Australia (Glen Forrest, Western Australia, Australia—dietary sheets can be found in Appendix A). Resistant starch (GemStar^®^ RS, RS type 4, at a dose of 110–140 g/Kg of food) was supplemented to the food.

Mice were humanely killed by cervical dislocation after 20 weeks. The small intestine and colon were immediately harvested and placed into 10% neutral buffered formalin overnight before subsequent histological processing and analysis. Distal colon was also collected for epithelial barrier integrity assessment.

### 2.2. Ussing Chamber

At the end of week 20, the segments of colon were cut open along the mesenteric border to expose the mucosa, washed with 1 X phosphate-buffered saline (Thermal Fisher Scientific, Scoresby, Australia) to remove luminal content, and mounted on P2311 Ussing Chamber sliders (0.3 cm^2^). Ussing sliders with tissue were inserted into two-part chambers (EasyMount Diffusion Chambers, Physiologic Instruments, Navicyte SDR Clinical Technology, Chatswood, Australia) containing physiological saline (Krebs) (115 mM NaCl, 25 mM NaHCO_3_, 2.4 mM K_2_HPO_4_, 1.2 mM CaCl_2_, 1.2 MgCl_2_, 0.4 mM KH_2_PO_4_, at pH 7.4) at 37 °C and were gassed with carbogen (5% CO_2_, 90% O_2_). Each chamber half contained 5 mL of the Krebs bicarbonate buffer, with the serosal bath having an additional 10 mM glucose and the mucosal bath containing an additional 10 mM mannitol. Each chamber had a set of 4 electrodes (2 voltage sensing and 2 current passing electrodes) which were installed on opposite sides of the tissue and connected to the amplifier through agar bridges and was bubbled with carbogen.

Short circuit current (Isc) was recorded using a multichannel voltage-current clamp (Physiologic Instruments, VCC MC6, Navicyte SDR Clinical Technology, Australia). Tissues were allowed to equilibrate in the chambers for 30 min before measurements were conducted. The tissues were then pulsed with voltages of 1, 2, 3, and 4 mV (positive and negative) for 3 s duration at each voltage and periods of 10 s in between voltages. Throughout the experiment, voltage and current readings were measured using Lab Chart software (ADinstruments, Bella Vista, Australia, version 8.1.13).

### 2.3. Haematoxylin and Eosin Staining

Small intestinal and colonic tissues were fixed in 10% neutral buffered formalin, processed, paraffin-embedded, and sectioned (4 μm) before staining with haematoxylin and eosin. Haematoxylin and eosin staining was performed using a standard protocol as previously described by Parlee et al. [75].

### 2.4. Imaging

Following staining, samples were examined and imaged using an EVOS^TM^ M7000 Imaging System (AMF7000, Thermal Fisher Scientific, Australia). Gut structural assessment was blindly assessed by randomising 10 out of 100 fields per gut section measuring changes in villi, crypt, goblet cells, and inflammatory cells at 20× magnification.

### 2.5. Histological Scoring

The histological scoring system was developed by a Deakin University pathologist (Deakin University, Australia) using Image Pro V6 software (Media Cybernetics, Rockville, MD, USA). Gut structural changes were examined using Image J software (version 1.52a, USA) following the scoring system listed in Table 1.

### 2.6. Statistical Analysis

Histological and epithelial barrier integrity analysis were statistically analysed using a 2-way ANOVA (diet and resistant starch status) with post hoc comparisons using Tukey’s honestly significant difference (HSD). The histological score of each mouse was determined by averaging the score of 10 random image fields. Analytical statistics were performed using GraphPad Prism (version 10.0.2). Data are presented as mean ± standard error of the mean (SEM), with a *p*-value of 0.05 used to determine significance.

## 3. Results

### 3.1. Small Intestine

#### 3.1.1. Inflammatory Cell Infiltration

After 20 weeks, in male mice, HF-fed mice had significantly increased inflammatory cell density in comparison to control-fed mice (2-way ANOVA, diet × RS status *p* < 0.0001, diet *p* < 0.0001, RS status *p* = 0.0626; Tukey’s HSD: *p* = 0.0021) (Figure 2A,C,G). Interestingly, yoyo mice appeared to be in an intermediate inflammatory state between control and HF states; however, this was not significantly different in comparison with control and HF groups (Tukey’s HSD: *p* = 0.2377 and *p* = 0.4141, respectively) (Figure 2A,C,E,G). In female mice, there was no significant difference in inflammatory cell density between control and HF-fed mice (2-way ANOVA, diet × RS status *p* = 0.0110, diet *p* = 0.0787, RS status *p* = 0.0138; Tukey’s HSD: *p* = 0.0774). In contrast, yoyo dieting resulted in significantly reduced inflammatory cell density in comparison to a control diet (Tukey’s HSD: *p* = 0.0063) (Figure 2G).

In male mice, RS supplemented in a control diet (control RS) significantly increased inflammatory cell density in comparison to a control diet alone (Tukey’s HSD: *p* = 0.0003) (Figure 2A,B,G). However, RS supplementation in HF-fed mice (HF RS) did not alter the inflammatory levels in comparison to a HF diet only (Tukey’s HSD: *p* = 0.4147) (Figure 2C,D,G). On the other hand, there was no difference in inflammatory cell density in control and HF-fed female mice supplemented with RS in comparison to those fed without RS (Tukey’s HSD: *p* = 0.9034 and *p* = 0.2876, respectively) (Figure 2G).

In male mice, RS supplementation in a yoyo diet (yoyo RS) resulted in significantly reduced inflammatory cell density in comparison to the yoyo group (Tukey’s HSD: *p* = 0.0174) (Figure 2E,F,G). Moreover, yoyo RS mice had significantly reduced inflammatory cell density in comparison to HF RS (Tukey’s HSD: *p* < 0.0001) (Figure 2D,F,G) and control RS mice (Tukey’s HSD: *p* < 0.0001) (Figure 2B,F,G). Conversely, in female mice, yoyo RS resulted in significantly elevated inflammatory cell density in comparison to the yoyo group (Tukey’s HSD: *p* = 0.0324), but not the HF RS group (Tukey’s HSD: *p* = 0.9999) (Figure 2G).

#### 3.1.2. Epithelial Change (Crypt Elongation)

From observation, the epithelial cells in small intestinal crypts across all diet groups appeared to be similar in size. Regardless of diet, sex, and RS status, there was no significant difference in crypt elongation ([male] 2-way ANOVA, diet × RS status *p* = 0.0542, diet *p* = 0.0125, RS status *p* = 0.1369; [female] 2-way ANOVA, diet × RS status *p* = 0.5954, diet *p* = 0.3925, RS status *p* = 0.0373) (Figure 3).

#### 3.1.3. Mucosal Architecture (Villous Loss)

From observation, regardless of sex, control-fed mice had intact villi with no signs of damage (Figure 4A and Figure 5A). On the other hand, HF-fed mice had shorter, swollen, and distorted villi with no signs of sloughing (Figure 4C and Figure 5C). We observed the most damage in the epithelial layer of male yoyo mice which was characterised by short, distorted villi, and signs of sloughing (Figure 4E). Moreover, the epithelial layer of groups supplemented with RS were mostly intact and did not show signs of damage (Figure 4 and Figure 5).

In male mice, HF-fed mice appear to have increased villous loss in comparison to control-fed mice; however, this was not significantly different (2-way ANOVA, diet × RS status *p* = 0.0176, diet *p* = 0.1130, RS status *p* = 0.0237; Tukey’s HSD: *p* = 0.4761) (Figure 4A,C,G). On the other hand, yoyo mice had significantly increased mucosal damage in comparison to control mice (Tukey’s HSD: *p* = 0.0102), and the same trend can be observed in HF-fed mice, although this was not significant (Tukey’s HSD: *p* = 0.4761) (Figure 4A,C,E,G). Similarly to males, HF-fed female mice had significantly increased villous loss in comparison to control-fed mice (2-way ANOVA, diet × RS status *p* = 0.0129, diet *p* = 0.0170, RS status *p* = 0.3832; Tukey’s HSD: *p* = 0.0017) (Figure 5A,C,G). Interestingly, there was no difference in mucosal architecture between female yoyo and control groups (Tukey’s HSD: *p* = 0.7622) (Figure 5A,E,G).

RS supplemented in a control diet (control RS) did not alter the mucosal architecture in male or female mice compared with a control diet only (Tukey’s HSD: [male] *p* = 0.9904, [female] *p* = 0.8284) (Figure 4A,B,G and Figure 5A,B,G). RS supplemented in a HF diet (HF RS) significantly reduced mucosal damage in comparison to a HF diet alone in female (Tukey’s HSD: *p* = 0.0461) (Figure 5C,D,G) but not in male mice (Tukey’s HSD: *p* = 0.8875) (Figure 4C,D,G).

RS supplementation in a yoyo diet (yoyo RS) resulted in significantly reduced villous loss in comparison to the yoyo group in male mice (Tukey’s HSD: *p* = 0.0102) (Figure 4E,F,G) but not in female mice (Tukey’s HSD: *p* > 0.9999) (Figure 5E,F,G). Moreover, there was no difference in the mucosal architecture between yoyo RS and HF RS in male and female mice (Tukey’s HSD: [male] *p* = 0.9850; [female] *p* = 0.9990) (Figure 4D,F,G and Figure 5D,F,G).

### 3.2. Colon

#### 3.2.1. Inflammatory Cell Infiltration

After 20 weeks, regardless of sex, there was no significant difference in inflammatory cell density between control and HF-fed mice ([male] 2-way ANOVA, diet × RS status *p* = 0.0002, diet *p* = 0.0017, RS status *p* = 0.0021, Tukey’s HSD: *p* = 0.9892; [female] 2-way ANOVA, diet × RS status *p* = 0.0019, diet *p* = 0.0017, RS status *p* = 0.0666, Tukey’s HSD: *p* = 0.7655) (Figure 6A,C,G). In contrast, yoyo mice had significantly increased inflammatory cell density in comparison to control male and female mice (Tukey’s HSD: [male] *p* < 0.0001; [female] *p* < 0.0001) (Figure 6A,E,G), and HF-fed male mice only (Tukey’s HSD: *p* = 0.0006) (Figure 6C,E,G).

Regardless of sex, RS supplemented in control diet (control RS) significantly increased inflammatory cell density in comparison to a control diet alone (Tukey’s HSD: [male] *p* = 0.0002, [female] *p* = 0.0014) (Figure 6A,B,G). In contrast, RS supplementation in HF mice (HF RS) did not alter inflammatory levels in comparison to mice fed a HF diet alone (Tukey’s HSD: [male] *p* = 0.2462, [female] *p* > 0.9999) (Figure 6C,D,G).

RS supplementation in a yoyo diet (yoyo RS) in male and female mice did not result in significantly different inflammatory cell levels in comparison to the yoyo group (Tukey’s HSD: [male] *p* = 0.6781, [female] *p* = 0.9769) (Figure 6E,F,G) and HF RS group (Tukey’s HSD: [male] *p* = 0.9743, [female] *p* = 0.9872) (Figure 6D,F,G).

#### 3.2.2. Epithelial Change (Crypt Elongation)

From observation, the colonic epithelium was mostly intact across all diet groups. However, crypt length in male control RS mice appeared to be increased in comparison to other groups (Figure 7).

After 20 weeks, there were no significant differences in crypt length in male mice without RS supplementation (2-way ANOVA, diet × RS status *p* = 0.0003, diet *p* = 0.4035, RS status *p* = 0.7687; Tukey’s HSD: [HF versus control] *p* = 0.3955; [HF versus yoyo] *p* > 0.9999; [yoyo versus control] *p* = 0.3955). Moreover, RS supplemented in a control diet (control RS) significantly increased crypt length in comparison with a control diet only in male mice (Tukey’s HSD: *p* = 0.0048). In contrast, RS supplemented in HF (HF RS) and yoyo diets (yoyo RS) resulted in significantly reduced crypt length in comparison to control RS (Tukey’s HSD: *p* = 0.0145 and *p* = 0.0094, respectively) (Figure 7).

In female mice, there were no significant differences in the epithelium across all diets (2-way ANOVA, diet × RS status *p* = 0.6770, diet *p* = 0.1605, RS status *p* = 0.1126) (Figure 7).

#### 3.2.3. Mucosal Architecture

Regardless of diet, sex, and RS status, there were no significant differences in colonic mucosal architecture in all dietary groups (all scored 0).

#### 3.2.4. Goblet Cell Change

From observation, goblet cell sizes and numbers were similar across most diet groups. Male mice consuming diets with control RS, HF, and HF RS appeared to have enlarged goblet cells. No difference in the morphology of goblet cells in female mice (Figure 8).

In male mice, yoyo dieting resulted in significantly increased goblet cells in comparison with a control diet (2-way ANOVA, diet × RS status *p* < 0.0001, diet *p* = 0.8978, RS status *p* = 0.0143, Tukey’s HSD: *p* = 0.0241). Yoyo dieting also appeared to have higher levels of goblet cells in comparison with continuous HF feeding, but the difference was not statistically significant (Tukey’s HSD: *p* = 0.6917). Supplementation with RS in a control diet (control RS) significantly increased goblet cell count in comparison with a control diet only (Tukey’s HSD: *p* = 0.0006). In contrast, RS supplementation in the HF group (HF RS) did not affect the number of goblet cells in comparison to a HF diet alone (Tukey’s HSD: *p* = 0.9999). Similarly, RS supplementation in a yoyo diet (yoyo RS) did not significantly alter the goblet cell count in comparison to yoyo (Tukey’s HSD: *p* = 0.2654) and HF RS groups (Tukey’s HSD: *p* = 0.9289) (Figure 8).

In female mice, there were no significant differences in goblet cell density across diets (2-way ANOVA, diet × RS status *p* = 0.8630, diet *p* = 0.2712, RS status *p* = 0.0449) (Figure 8).

### 3.3. Epithelial Barrier Integrity

Regardless of sex and RS status, there were no significant differences in colonic epithelial barrier integrity across all diets ([male] 2-way ANOVA, diet × RS status *p* = 0.8664, diet *p* = 0.0427, RS status *p* = 0.3351; [female] 2-way ANOVA, diet × RS status *p* = 0.2006, diet *p* = 0.2596, RS status *p* = 0.0449) (Figure 9).

## 4. Discussion

There are five different types of RS that can be classified based on chemical composition, structure, and source [56]. Previous studies have mostly focused on the effects of RS2, which can commonly be found in high-amylose maize starch, green banana, and raw potato [65,76,77,78,79,80,81,82,83,84,85,86,87]. While RS2 is more commonly found in foods, they are less resistant and easily degraded by heat [88]. Another type of RS that is chemically modified, RS4, is commercially available in baked goods [89,90] (such as cookies [90], cakes [91], and breads [92]) and has been shown to offer greater chemical resistance and health benefits in comparison with RS2 [88]. Hence, we utilised RS4 to supplement dietary treatments in this study. To date, this is the first study to investigate the effect of yoyo dieting and RS4 on gastrointestinal structures. Our findings revealed that yoyo dieting worsened gut inflammation compared to continuous high fat diet feeding. We also showed that supplementation with RS appears to be both beneficial and detrimental: while improving mucosal barrier integrity, it also exacerbated intestinal inflammation in the absence of a dietary challenge.

It is well established that the gut is one of the most susceptible organs responding to diets [93]. High fat diet feeding has been linked to gut dysbiosis [94,95,96,97], increased gut permeability [98,99,100], gut inflammation [101,102,103], and inflammatory bowel disease development [104,105,106], of which, increased gastrointestinal inflammation and damaged mucosal architecture are key features [107]. Frequent changes in diet composition (alternating high fat and control diets) during yoyo dieting may lead to constant fluctuations in the intestinal environment. Emerging evidence suggests that yoyo dieting is associated with heightened systemic inflammation [108,109], which is consistent with the increased colonic inflammation that we observed in yoyo dieting. Moreover, increased mucosal damage, characterised by short and distorted villi, with signs of accelerated sloughing, has been shown in high fat feeding [98,110,111,112]. This is in line with the mucosal damage that we observed in our high fat and yoyo mice. While there have been no previous studies that have directly investigated gut morphological changes following yoyo dieting, increasing evidence suggests that yoyo dieting is associated with gut dysbiosis, such as reduced *Christensenella* spp. and *Lactobacillus reuteri* [113,114] and enriched *Ruminococcus* [115]. It is likely that the reduction in *Christensenella* spp. and *Lactobacillus reuteri* will be detrimental as these bacteria have been related to a healthy phenotype, with reductions being associated with increased inflammation [116], reduced intestinal barrier function [117], and increased risk of developing inflammatory bowel disease [118]. Similarly, enriched *Ruminococcus* has been implicated in Crohn’s disease [119] and metabolic syndrome [120]. It has also been shown that yoyo dieting induces long-term alterations in the gut microbiome composition [121] and these changes persist even after returning to normal weight [37,121]. Our findings provide novel evidence to show that mucosal damage and colonic inflammation are exacerbated in yoyo dieting. When the mucosal barrier is compromised, the gut becomes permeable, triggering the immune system and causing inflammation which may be conducive to the long-term perturbations of the gut microbiota that have been observed in the literature. However, it is also likely that the gut dysbiosis associated with yoyo dieting drives the exacerbation of mucosal damage and inflammation. Further investigations are required to elucidate this complex interplay between diet, gastrointestinal barrier integrity, inflammation, and microbiome.

There is increasing and compelling evidence around the beneficial effects of RS on improving gut barrier integrity [122,123]. In particular, Kadyan et al. [122] explored the effect of RS on 60-week-old humanised mice fed a high fat diet for 20 weeks. It was shown that RS supplemented in a high fat diet significantly increased the expression of tight junction proteins (claudin-1, claudin-4, occludin, and zonulin-2) and reduced gut leakiness [122]. Similarly, a recent study by Li et al. [79] found increased expression of tight junction proteins (zonulin-1 and occludin) in 12-week-old mice following faecal microbiota transplantation from obese human donors following RS intervention. RS supplementation in 12-week-old male mice has also been shown to ameliorate the gut barrier dysfunction induced by high fat feeding [123]. Consistent with the literature, we observed improved mucosal structure following RS supplementation in high fat and yoyo groups. In addition to improving the gut barrier, RS has also been shown to reduce inflammation in various diseases (obesity, prediabetes/diabetes, and chronic kidney disease) [79,87,124,125,126,127,128]. In contrast to this anti-inflammatory effect, we found that RS supplementation did not reduce inflammation in high fat-fed mice, which is corroborated by other studies [129,130]. Additionally, we provide novel evidence that RS supplementation in a control diet led to increased intestinal inflammation and augmented crypt depth. It is plausible that these changes may contribute to gastrointestinal symptoms (discomfort, bloating, and alterations in bowel habit), and reduction in nutrient absorption commonly associated with high RS consumption [131,132]. Of note, high dietary fibre consumption has also been positively correlated with significant enrichment in several gut microbiome [133,134,135,136], such as *Prevotella* [133,134,135,136], *Eggerthella*, and Lachnospiraceae [137]. Increased *Prevotella* has been implicated in inflammatory bowel disease [138,139,140,141], HIV [142,143,144], and increased gut permeability [142,143,144]. Similarly, elevated levels of *Eggerthella* and Lachnospiraceae have been associated with pro-inflammatory effects in individuals with inflammatory bowel disease [145], rheumatoid arthritis [146], and diabetes [147,148,149].

Moreover, we also observed that yoyo dieting appeared to have a more pronounced negative effect on gut morphology in male compared to female mice. This is consistent with previous studies suggesting that a high fat diet is a risk factor for long-term gastrointestinal issues; however, most studies focus predominantly on males [150,151]. Given the sex differences we observed, our study underscores the importance of investigating both sexes in obesity and yoyo dieting, given the limited number of studies addressing this issue.

While this study offered novel insights into the deleterious effects of yoyo dieting and RS supplementation on gastrointestinal structure, several limitations need to be considered. Although we did not observe changes in epithelial barrier function, we only assessed the colon and there may be changes in other gut regions. Moreover, we only analysed inflammation histologically and inflammatory biomarkers were not measured. Similarly, histological assessment could only be performed at the experimental endpoint. Therefore, potential histological changes when switching between diets, which is an important feature of yoyo dieting, could not be evaluated. Future directions in this area of research should include investigations in gut microbiota changes, as well as clinical studies utilising RS4 supplementation.

## 5. Conclusions

To date, this is the first study to investigate the relationship between gastrointestinal structure, yoyo dieting, and resistant starch supplementation. This investigation highlights that frequent on–off dieting (yoyo) exacerbates gut inflammation even more than chronic obesity. Resistant starch also appears to be a double-edged sword, improving mucosal barrier integrity while also exacerbating intestinal inflammation in the absence of a dietary challenge. These findings are crucial for supporting the development of evidence-based lifestyle interventions, such as increased intake of resistant starch, to improve gut health. Further research, particularly in humans, is required to investigate the long-term effects and optimal dosage of resistant starch supplementation, as well as to gain a deeper understanding of the mechanisms through which yoyo dieting impacts gut health.

## Figures and Tables

**Figure 1 nutrients-16-04216-f001:**
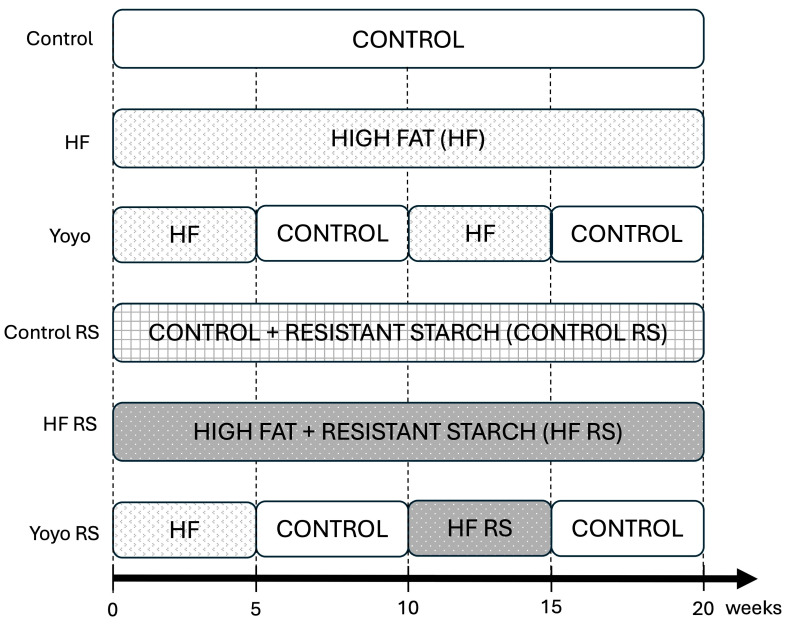
20-week dietary treatments.

**Figure 2 nutrients-16-04216-f002:**
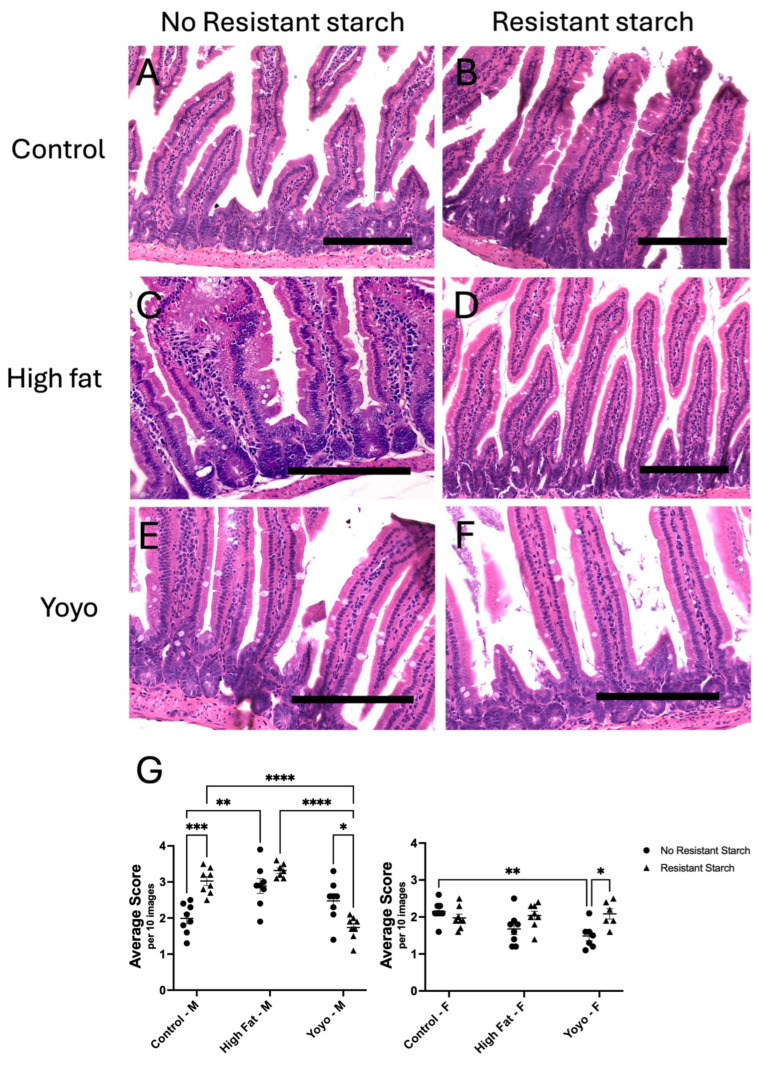
Small intestinal analysis of inflammatory cell density. Images of inflammatory cell density in male mice (**A**–**F**). In male mice, inflammatory cell density of (**C**) HF mice is greater than (**A**) control mice. Inflammatory cell count of (**E**) yoyo mice appears to be in an intermediate state between (**A**) control and (**C**) HF states. Higher inflammatory cell count is observed in (**B**) control RS mice in comparison to (**A**) control mice. Inflammatory cell density of (**D**) HF RS mice is increased in comparison to (**C**) HF mice. Inflammatory cell count of (**F**) yoyo RS mice is lower than (**E**) yoyo mice. Statistical analysis of changes in the inflammatory cell density of male and female mice (**G**), with * *p* ≤ 0.05, ** *p* ≤ 0.01, *** *p* ≤ 0.001, **** *p* ≤ 0.0001 used to determine significance. Scale bar = 150 µm. M: male; F: female.

**Figure 3 nutrients-16-04216-f003:**
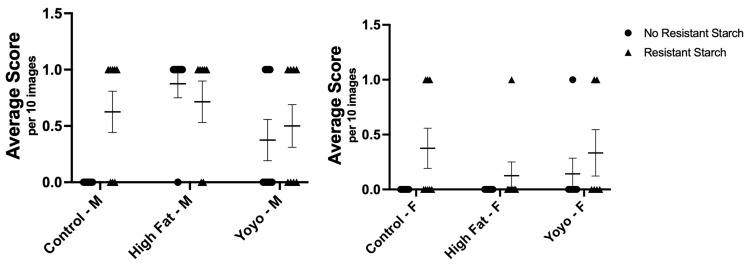
Small intestinal analysis of epithelial change. The epithelial cells in small intestinal crypts across all diet groups appeared to be similar in size. No significant differences were observed between all groups. M: male; F: female.

**Figure 4 nutrients-16-04216-f004:**
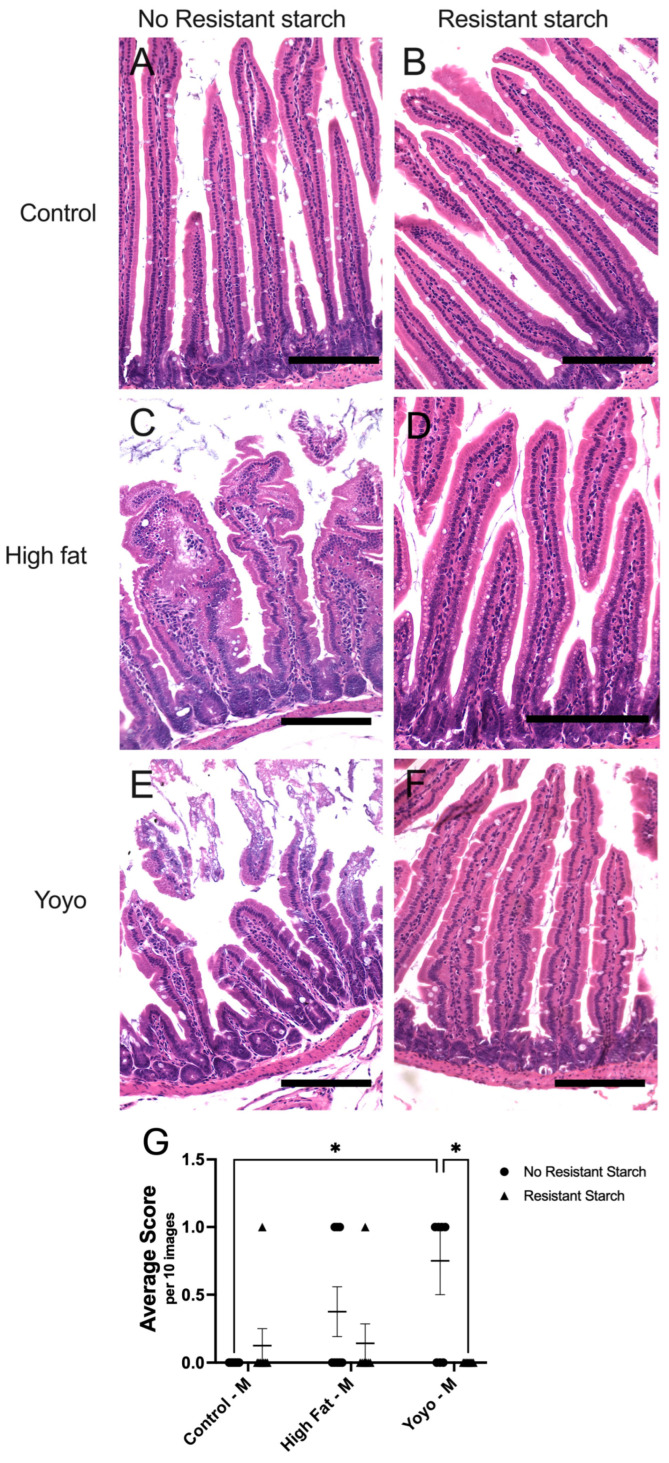
Small intestinal analysis of mucosal changes in male mice. Intact mucosa in (**A**) control and (**B**) control RS mice. Swollen and distorted villi in (**C**) HF mice. Intact mucosa in (**D**) HF RS mice. Swollen, distorted and damaged mucosa in (**E**) yoyo mice but intact mucosa in (**F**) yoyo RS mice. Statistical analysis of changes in the mucosal architecture of male mice (**G**), with * *p* ≤ 0.05 used to determine significance. Scale bar = 150 µm. M: male.

**Figure 5 nutrients-16-04216-f005:**
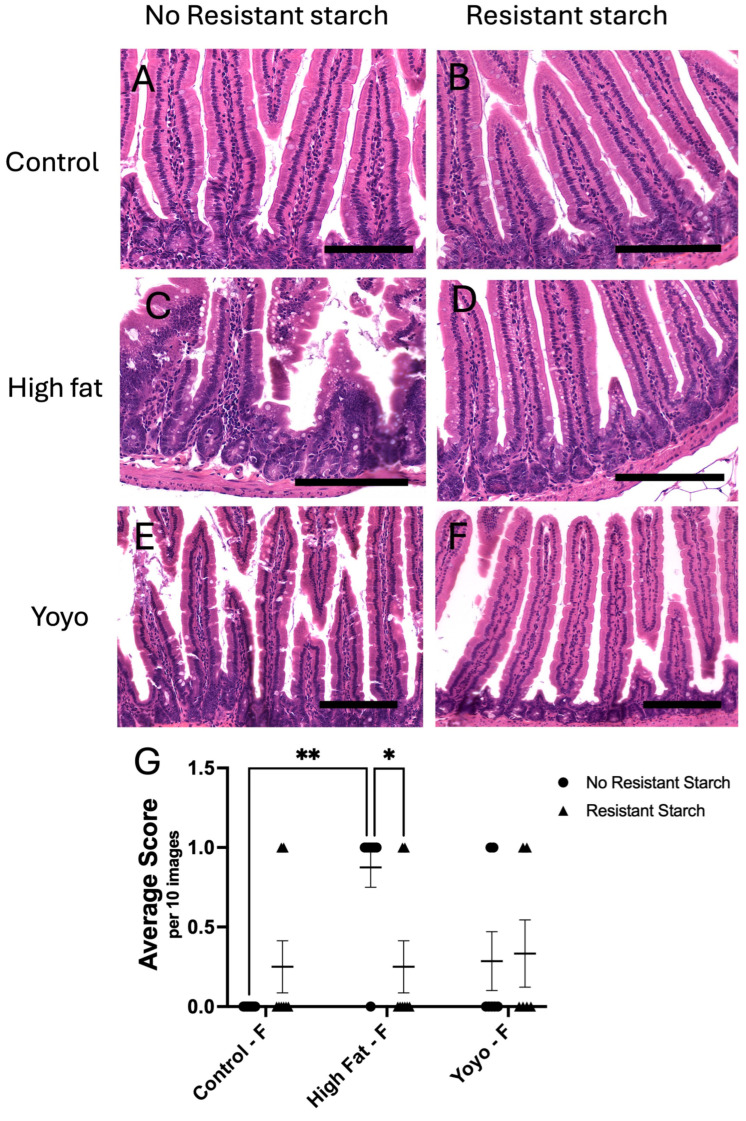
Small intestinal analysis of mucosal changes in female mice. Mucosal damage was observed only in (**C**) HF mice, with intact mucosa in (**A**) control, (**B**) control RS, (**D**) HF RS, (**E**) yoyo, and (**F**) yoyo RS mice. Statistical analysis of changes in the mucosal architecture of female mice (**G**), with * *p* ≤ 0.05, ** *p* ≤ 0.01 used to determine significance. Scale bar = 150 µm. F: female.

**Figure 6 nutrients-16-04216-f006:**
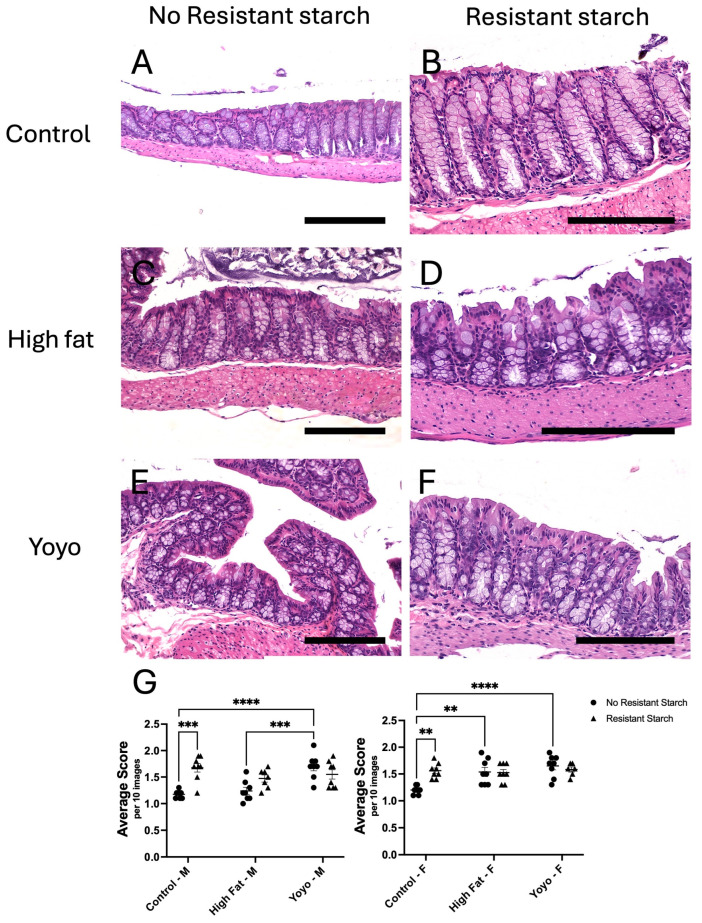
Colonic analysis of inflammatory cell density. Images of inflammatory cell density in male mice (**A**–**F**). Regardless of sex, inflammatory cell density of (**B**) control RS mice is higher than (**A**) control mice. In male mice, inflammatory cell density of (**E**) yoyo mice is higher than (**C**) HF and (**A**) control mice. Inflammatory cell density of (**B**) control RS and (**F**) yoyo mice are similar. Inflammatory cell density of (**D**) HF RS and (**F**) yoyo RS mice do not differ and are lower than (**B**) control RS mice. In female mice, the inflammatory cell density of control mice is lower than other groups. Statistical analysis of changes in colonic inflammatory cell density of male and female mice (**G**), with ** *p* ≤ 0.01, *** *p* ≤ 0.001, **** *p* ≤ 0.0001 used to determine significance. Scale bar = 150 µm. M: male; F: female.

**Figure 7 nutrients-16-04216-f007:**
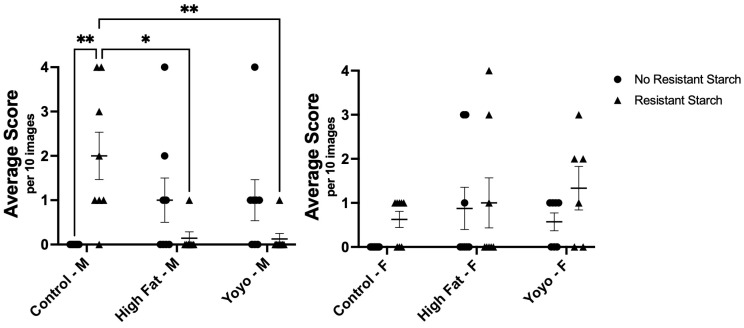
Histological analysis of epithelial change in the colon of male and female mice. Male control RS mice had increased crypt length in comparison to control, HF RS, and yoyo RS mice. Male HF and yoyo mice had similar crypt lengths. No significant differences in epithelial change were observed in female mice. * *p* ≤ 0.05, ** *p* ≤ 0.01 used to determine significance. M: male; F: female.

**Figure 8 nutrients-16-04216-f008:**
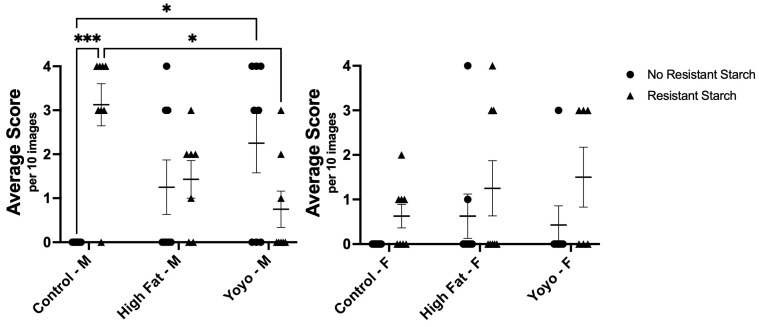
Histological scores of goblet cell change. Male control RS mice had significantly higher goblet cell density compared to control and yoyo RS mice. Male yoyo mice had increased goblet cell density in comparison with control mice. No significant differences were observed in other groups and in female mice. * *p* ≤ 0.05, *** *p* ≤ 0.001 used to determine significance. M: male; F: female.

**Figure 9 nutrients-16-04216-f009:**
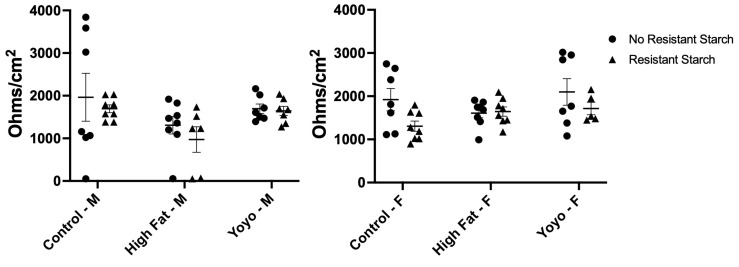
Colonic epithelial barrier integrity measured in Ussing chambers. No significant differences were observed between all groups. M: male; F: female.

**Table 1 nutrients-16-04216-t001:** Histological scoring criteria for (**a**) small intestine and (**b**) colon.

**(a)** **Small intestine**		
**Category**	**Definition**	**Score**
Inflammatory cell infiltration	Increased inflammatory cell density of lamina propria involving villi and crypts in comparison to controls	
	Minimal: <10%	1
	Mild: 10–25%	2
	Moderate: 26–50%	3
	Marked: >51%, dense infiltrate	4
Epithelial change	Increased epithelial cell numbers in longitudinal crypts in comparison to controls	
	Minimal: <25%	1
	Mild: 25–35%	2
	Moderate: 35–50%; mitoses in the middle/upper third of crypt epithelium	3
	Marked: >51%, mitoses in upper third of crypt epithelium	4
Mucosal architecture	Decrease in the epithelium: from simple erosion to complete loss of epithelium in comparison to controls	
	Mild: villous loss 1/2 length to normal	1
	Moderate: villous loss 2/3 length to normal	2
	Severe: mucosal devoid of villi	3
	Ulceration: epithelial loss is so severe that is up to the deeper layer of muscularis	4
**(b)** **Colon**		
**Category**	**Definition**	**Score**
Inflammatory cell infiltration	Leukocyte density of lamina propria involving crypts in comparison to controls	
	Minimal: <10%	1
	Mild: 10–25%	2
	Moderate: 26–50%	3
	Marked: >51%, dense infiltrate	4
Epithelial change	Increase in epithelial cell numbers in longitudinal crypts in comparison to controls	
	Minimal: <25%	1
	Mild: 25–35%	2
	Moderate: 35–50%; mitoses in the middle/upper third of crypt epithelium	3
	Marked: >51%, mitoses in upper third of crypt epithelium	4
Mucosal architecture	Decrease in depth of crypts in comparison to controls	
	Mild: crypt 1/2 length to normal	1
	Moderate: crypt 2/3 length to normal	2
	Severe: mucosal devoid of crypt	3
	Ulceration: epithelial loss is so severe that is up to the deeper layer of muscularis	4
Goblet cell changes	Hyperplasia (increased number of goblet cell per crypt) in comparison to controls	
	Minimal: <10%	1
	Mild: 10–25%	2
	Moderate: 26–50%	3
	Marked: >51%	4

## Data Availability

The original contributions presented in this study are included in the article. Further inquiries can be directed to the corresponding author.

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
