# Peer review of "Deleterious Effects of Yoyo Dieting and Resistant Starch on Gastrointestinal Morphology"

_nutrients, 2024, doi:10.3390/nu16234216_

Round 1
Reviewer 1 Report
Comments and Suggestions for Authors
In this study, the effect of yoyo dieting on gastrointestinal structure and whether resistant starch (RS) has beneficial effects in improving obesity–related gastrointestinal damage were investigated. C57BL/6 mice were assigned to 6 different diets for 20 weeks, including: (1) control diet, (2) high fat diet (HF), (3) yoyo diet (alternating HF and control diets every 5 weeks), (4) control diet with RS, (5) HF with RS, and (6) yoyo diet with RS. Distal colon was collected for epithelial barrier integrity measurement. The small and large intestines were collected for histological assessment. At the end of the dietary intervention, yoyo dieting resulted in increased colonic inflammation and exacerbated mucosal damage compared to continuous HF diet feeding. This manuscript brought some new knowledge to reader. The topic of this article is intriguing and fully fills in the scope of Nutrients. However, there are some questions in this manuscript. For these reasons, I think it should be acceptable for publication after revision. The following are the comments about this manuscript:
1. There isn’t an abbreviation list. Some undefined abbreviations in the manuscript are confusing to the readers.
2. Keywords, the keyword is too long, which should be more concise. 3. In Materials section, the authors should introduce the chemical composition, structure, and source of RS.
4. Why high-fat mice model was selected?
5. Line 106, “MgCl2” should be “MgCl2”.
Author Response
Dear reviewer,
Thank you for giving us the opportunity to submit a revised draft of our manuscript titled ‘Deleterious effects of yoyo dieting and resistant starch on gastrointestinal morphology’ to Nutrients. We appreciate the time and effort that you have dedicated to providing valuable and insightful feedback on our manuscript. We have addressed your comments (please see below) and the manuscript is now improved. Sentences highlighted in blue are responses addressing the reviewer’s comments and track changes that have also been made within the manuscript.
- There isn’t an abbreviation list. Some undefined abbreviations in the manuscript are confusing to the readers.
Thank you for your suggestion. The abbreviation list is not a requirement in the submission guidelines by Nutrients. However, we have thoroughly checked and ensured that all abbreviations were defined in the manuscript.
- Keywords, the keyword is too long, which should be more concise.
This has been amended to make the keyword list only consist of 5 keywords.
- In Materials section, the authors should introduce the chemical composition, structure, and source of RS.
All diets, including RS supplementation (GemStar® RS) used in this study, were from Specialty Feeds Australia (Glen Forest, Western Australia) (section 2.1, lines 87-88). Dietary sheets were also submitted as Supplementary Material.
- Why high-fat mice model was selected?
The aim of this study is to investigate the effects of yoyo dieting (HF diet interspaced with control diet) and resistant starch (RS) supplementation on gastrointestinal morphology, and high fat feeding was required in order to induce yoyo dieting.
- Line 106, “MgCl2” should be “MgCl2”.
This has been amended.
Thank you once again for your feedback on how to improve this manuscript. We look forward to hearing from you in due time regarding our submission and to responding to any further questions and comments you may have.
Sincerely,
Kate Phuong–Nguyen, Malik Mahmood, and Leni Rivera
Reviewer 2 Report
Comments and Suggestions for Authors
General:
The study by Kate Phuong-Nguyen et al., with the title “Deleterious effects of yoyo dieting and resistant starch on gastrointestinal morphology” focuses on studying whether different diet such as resistant starch may be beneficial in promoting structural improvements in the gut. The paper is interesting, but I have the following question.
I have the following comments.
1. Physiological saline (page 4, lane 105)? Is this the same solution as Krebs solution that you mention later down (page 4, lane 109)? You also mention that you added glucose and mannitol to maintain osmotic balance twice in this section.
2. Did you wash the intestinal before you put it in the Ussing chamber? If you have some food left in the intestine, it affects the measurement in the Ussing chamber a lot.
3. In the Ussing chamber, you also measured the voltage (potential difference). I miss the result on this.
4. In Figure 2 and 6 you describe that you analys “inflammatory cell density”, but in Figure 3, 4, 5 and 7 you did not write that you analyse in figure text. Please print.
5. The text in Figure 6 is wrong “inflammatory cell density in male mice”. You have analysed female mice too.
6. The discussion first lacks a summary of the results. It is needed as the result part is quite complex. Why did you divide the animals into male and female groups? Were there gender differences? It cannot be seen in the graphs. Is it possible to make a summary for the different genders and then a total summary? And of course, same discussion about that.
7. I lack information about the mice after 20 weeks of diet. How did they feel, and did they gain or lose weight depending on the diet?
8. You discuss a lot about different kinds of bacteria being either protective or harmful to the gut. But why didn´t you analyze the bacteria that were present in the gut after 20 weeks of different diets?
9. In your conclusion, you write that further studies are needed, especially on humans, to see the effect of different diet over a longer period of time. Do you think that 20 weeks diet on mice is a long time or a short time?
Animal experiments and ARRIVE checklist:
The checklist is almost complete, but I am missing the following points:
Nr 16.a. What is meant by “Animals were”?
Nr 16.c. What is meant by “Animals were”?
Nr 21.a. An answer is missing here.
Nr 21.b. An answer is missing here.
Author Response
Dear reviewer,
Thank you for giving us the opportunity to submit a revised draft of our manuscript titled ‘Deleterious effects of yoyo dieting and resistant starch on gastrointestinal morphology’ to Nutrients. We appreciate the time and effort that you have dedicated to providing valuable and insightful feedback on our manuscript. We have addressed your comments (please see below) and the manuscript is now improved. Sentences highlighted in blue are responses addressing the reviewer’s comments and track changes that have also been made within the manuscript.
The study by Kate Phuong-Nguyen et al., with the title “Deleterious effects of yoyo dieting and resistant starch on gastrointestinal morphology” focuses on studying whether different diet such as resistant starch may be beneficial in promoting structural improvements in the gut. The paper is interesting, but I have the following question.
I have the following comments.
-
- Physiological saline (page 4, lane 105)? Is this the same solution as Krebs solution that you mention later down (page 4, lane 109)?
Yes, physiological saline is also called Krebs solution, which is made up of 115 mM NaCl, 25 mM NaHCO3, 2.4 mM K2HPO4, 1.2 mM CaCl2, 1.2 MgCl2, 0.4 mM KH2PO4, pH 7.4 (lines 293-294). To add more clarity, we have added “(Krebs)” in line 293.
You also mention that you added glucose and mannitol to maintain osmotic balance twice in this section.
Thank you, the duplication has been removed.
- Did you wash the intestinal before you put it in the Ussing chamber? If you have some food left in the intestine, it affects the measurement in the Ussing chamber a lot.
Segments of the distal colon were cut open along the mesenteric border to expose the mucosa (lines 288-289) and washed with 1X phosphate buffered saline (PBS) to remove luminal contents. Additional information “washed with 1X phosphate buffered saline (Thermo Fisher Scientific, Australia) to remove luminal content” was added to lines 289-290.
- In the Ussing chamber, you also measured the voltage (potential difference). I miss the result on this.
Thank you for your comment. We have reported the transepithelial resistance (ohms/cm2) as a measure of epithelial barrier integrity. In this system, transmucosal voltage is stepped to four graded levels under voltage clamp conditions (clamped to 0mv), and the corresponding currents were measured. Transepithelial electrical resistance was then calculated by Ohm’s law and multiplied by the exposed area.
- In Figure 2 and 6 you describe that you analyse “inflammatory cell density”, but in Figure 3, 4, 5 and 7 you did not write that you analyse in figure text. Please print.
Thank you for your comment. This information has been listed in the figure titles:
Figure 3: Small intestinal analysis of epithelial change (line 506)
Figure 4: Small intestinal analysis of mucosal changes in male mice (line 557)
Figure 5: Small intestinal analysis of mucosal changes in female mice (line 565)
Figure 7: Histological analysis of epithelial change in the colon of male and female mice. (line 633)
- The text in Figure 6 is wrong “inflammatory cell density in male mice”. You have analysed female mice too.
Thank you for your comment. This has been amended by removing “in male mice” in the title of Figure 6 (line 606).
- The discussion first lacks a summary of the results. It is needed as the result part is quite complex. Why did you divide the animals into male and female groups? Were there gender differences? It cannot be seen in the graphs. Is it possible to make a summary for the different genders and then a total summary? And of course, same discussion about that.
Thank you for your comment. A summary of the results has been added as the first paragraph of the discussion section: “Our findings revealed that yoyo dieting worsened gut inflammation compared to continuous HF diet feeding. We also showed that supplementation with RS appears to be both beneficial and detrimental: while improving mucosal barrier integrity, it also exacerbated intestinal inflammation in the absence of a dietary challenge.” (lines 713-716)
We found sex differences in our study's results, with females showing greater protection against gastrointestinal disturbances compared to males, potentially due to different gut microbiome signatures between sexes. Discussion on sex differences has been added “Moreover, we also observed that yoyo dieting appeared to have a more pronounced negative effect on gut morphology in male compared to female mice. This is consistent with previous studies suggesting that high fat diet is a risk factor for long–term gastrointestinal issues, however, most studies focus predominantly on males [150, 151]. Given the sex differences we observed, our study underscores the importance of investigating both sexes in obesity and yoyo dieting, given the limited number of studies addressing this issue.” (lines 783-789).
- I lack information about the mice after 20 weeks of diet. How did they feel, and did they gain or lose weight depending on the diet?
We acknowledge that body weight change is important, and our previously published article focused on yoyo dieting, metabolism and gut microbiome (Phuong-Nguyen et al., 2024) reported changes in body weight and metabolic measures after 20 weeks of dietary challenge. In this study, after 20 weeks, we showed that yoyo mice had similar body weight change and fat mass compared to control mice. We also showed that resistant starch supplementation in a high fat diet significantly reduced fat mass and body weight in male mice compared to those fed with high fat diet only. This data has not been included in this paper as the focus is on gastrointestinal morphology.
- You discuss a lot about different kinds of bacteria being either protective or harmful to the gut. But why didn’t you analyze the bacteria that were present in the gut after 20 weeks of different diets?
Our previously published article focused on yoyo dieting, metabolism and gut microbiome (Phuong-Nguyen et al., 2024) reported changes in gut microbiome during and after 20 weeks of dietary challenge. Data of gut microbiome has not been included in this paper as the focus is on gastrointestinal morphology.
- In your conclusion, you write that further studies are needed, especially on humans, to see the effect of different diet over a longer period of time. Do you think that 20 weeks diet on mice is a long time or a short time?
While there have been conflicting findings regarding the optimal timing for dietary length affecting behavioural interventions, the literature suggests that most intervention periods are often advised to be a minimum of 6–8 weeks to form a habit (Feil et al., 2021). However, habit strength requires a minimum of 28 weeks (Fournier et al., 2017) which is much longer than 5–10 weeks of intervention timelines. Therefore, the timespan of our animal study is reflective of a longer equivalent period in humans (with mice aged 6–26 weeks corresponding to approximately humans aged 18–32 years old (Flurkey et al., 2007, Fox et al., 2007)). Hence, in our conclusion, we suggested that more studies, especially in humans, are warranted to explore the long–term implications and effective dosage of resistant starch supplementation and elucidate the underlying mechanisms by which yoyo dieting affects gut health.
Animal experiments and ARRIVE checklist:
The checklist is almost complete, but I am missing the following points:
Nr 16.a. What is meant by “Animals were”?
Animals were humanely culled from the study if they showed clinical illness or experienced 20% body weight loss over 7 days.
Nr 16.c. What is meant by “Animals were”?
Animals were monitored daily during the study.
Nr 21.a. An answer is missing here.
The authors declare no conflicts of interest (Line 860)
Nr 21.b. An answer is missing here.
This research received no external funding (line 853)
Thank you once again for your feedback on how to improve this manuscript. We look forward to hearing from you in due time regarding our submission and to responding to any further questions and comments you may have.
Sincerely,
Kate Phuong–Nguyen, Malik Mahmood, and Leni Rivera
References
CHEN, J., CHEN, L., SANSEAU, P., FREUDENBERG, J. M. & RAJPAL, D. K. 2016. Significant obesity-associated gene expression changes occur in the stomach but not intestines in obese mice. Physiological Reports, 4, e12793.
FEIL, K., ALLION, S., WEYLAND, S. & JEKAUC, D. 2021. A Systematic Review Examining the Relationship Between Habit and Physical Activity Behavior in Longitudinal Studies. Front Psychol, 12, 626750.
FLURKEY, K., M. CURRER, J. & HARRISON, D. E. 2007. Chapter 20 - Mouse Models in Aging Research. In:FOX, J. G., DAVISSON, M. T., QUIMBY, F. W., BARTHOLD, S. W., NEWCOMER, C. E. & SMITH, A. L. (eds.) The Mouse in Biomedical Research (Second Edition). Burlington: Academic Press.
FOURNIER, M., D'ARRIPE‐LONGUEVILLE, F. & RADEL, R. 2017. Testing the effect of text messaging cues to promote physical activity habits: a worksite‐based exploratory intervention. Scandinavian journal of medicine & science in sports, 27, 1157-1165.
FOX, J., BARTHOLD, S. W., DAVISSON, M. T., NEWCOMER, C. E., QUIMBY, F. & SMITH, A. L. 2007. The Mouse in Biomedical Research.
GARIDOU, L., POMIé, C., KLOPP, P., WAGET, A., CHARPENTIER, J., ALOULOU, M., GIRY, A., SERINO, M., STENMAN, L., LAHTINEN, S., DRAY, C., IACOVONI, JASON S., COURTNEY, M., COLLET, X., AMAR, J., SERVANT, F., LELOUVIER, B., VALET, P., EBERL, G., FAZILLEAU, N., DOUIN-ECHINARD, V., HEYMES, C. & BURCELIN, R. 2015. The Gut Microbiota Regulates Intestinal CD4 T Cells Expressing RORγt and Controls Metabolic Disease. Cell Metabolism, 22, 100-112.
PHUONG-NGUYEN, K., O’HELY, M., KOWALSKI, G. M., MCGEE, S. L., ASTON-MOURNEY, K., CONNOR, T., MAHMOOD, M. Q. & RIVERA, L. R. 2024. The Impact of Yoyo Dieting and Resistant Starch on Weight Loss and Gut Microbiome in C57Bl/6 Mice. Nutrients, 16, 3138.